# Early Detection of Both *Pyrenophora teres* f. *teres* and f. *maculata* in Asymptomatic Barley Leaves Using Digital Droplet PCR (ddPCR)

**DOI:** 10.3390/ijms252211980

**Published:** 2024-11-07

**Authors:** Yassine Bouhouch, Dina Aggad, Nicolas Richet, Sajid Rehman, Muamar Al-Jaboobi, Zakaria Kehel, Qassim Esmaeel, Majida Hafidi, Cédric Jacquard, Lisa Sanchez

**Affiliations:** 1INRAE, RIBP, Université de Reims Champagne-Ardenne, USC 1488, BP 1039 Reims, France; yassine.bouhouch@univ-reims.fr (Y.B.); richetnicolas@free.fr (N.R.); qassim.esmaeel@univ-reims.fr (Q.E.); cedric.jacquard@univ-reims.fr (C.J.); 2Plateformes Technologiques URCATech, Plateau MOBICYTE, Université de Reims Champagne-Ardenne, BP 1039 Reims, France; dina.aggad@univ-reims.fr; 3Biodiversity and Crop Improvement Program, International Center for Agricultural Research in the Dry Areas, Rabat BP 6202, Morocco; srehman@oldscollege.ca (S.R.); ljaboobimuamar@gmail.com (M.A.-J.); z.kehel@cgiar.org (Z.K.); 4Laboratoire de Biotechnologie Végétale et de Biologie Moléculaire, Faculté des Sciences, Université Moulay Ismail, Zitoune, Meknès BP 11201, Morocco; hafidimaj@yahoo.fr

**Keywords:** barley, *Pyrenophora teres*, k-mer analysis, qPCR, ddPCR, early detection

## Abstract

Efficient early pathogen detection, before symptom apparition, is crucial for optimizing disease management. In barley, the fungal pathogen *Pyrenophora teres* is the causative agent of net blotch disease, which exists in two forms: *P. teres* f. sp. *teres* (*Ptt*), causing net-form of net blotch (NTNB), and *P. teres* f. sp. *maculata* (*Ptm*), responsible for spot-form of net blotch (STNB). In this study, we developed primers and a TaqMan probe to detect both *Ptt* and *Ptm*. A comprehensive k-mer based analysis was performed across a collection of *P. teres* genomes to identify the conserved regions that had potential as universal genetic markers. These regions were then analyzed for their prevalence and copy number across diverse Moroccan *P. teres* strains, using both a k-mer analysis for sequence identification and a phylogenetic assessment to establish genetic relatedness. The designed primer-probe set was successfully validated through qPCR, and early disease detection, prior to symptom development, was achieved using ddPCR. The k-mer analysis performed across the available *P. teres* genomes suggests the potential for these sequences to serve as universal markers for *P. teres*, transcending environmental variations.

## 1. Introduction

Barley (*Hordeum vulgare* L.), belonging to the *Poaceae* grass family, is the fourth most important cereal crop globally in terms of yield and cultivated area, following wheat, rice, and maize [1]. Like other plants, barley must contend with attacks from numerous pathogens like *Pyrenophora teres*, *Pyrenophora graminea*, *Fusarium* sp., *Rynchosporium commune*, *Puccinia hordei*, and *Cochliobolus sativus* [2]. Among the diseases affecting the barley, net blotch can reduce yields by up to 40% and affect both seed quality and quantity [3]. This foliar disease is caused by the ascomycete fungus *Pyrenophora teres* and exists in two different forms: net-form of net blotch (NTNB), caused by *P. teres* f. sp. *teres* (*Ptt*), and spot-form of net blotch (STNB), caused by *P. teres* f. sp. *maculata* (*Ptm*) [4,5]. The net type is characterized by typical dark brown blotches that extend along the leaf axis with occasional rectangle striations, whereas the spot form causes oval-shaped dark brown leaf lesions [3,6].

In the field of genomics, k-mer analysis has become a vital tool for exploring strain diversity and understanding copy number variations. K-mers, defined as subsequences of length ‘k’ within a DNA sequence, offer insights into genetic diversity and the evolutionary processes underlying genomic changes [7,8]. One significant application of k-mer analysis is assessing changes in the copy number of highly repetitive sequences, which are often observed in complex genomes like those of plants. Two studies in maize revealed that the divergence in copy number among highly repetitive DNA sequences can be efficiently analyzed using k-mer spectra, highlighting their role in genomic evolution and domestication processes [9,10]. In the context of gene copy number estimation, tools like GeneToCN, which use an alignment-free method directly from next-generation sequencing reads, have proven to be effective. Such methods have been validated using ddPCR, highlighting the compatibility of k-mer analysis with contemporary genomic technologies [11]. Moreover, k-mer-based approaches, like PanKmer, offer a reference-free analysis of pangenomes. This tool can calculate pairwise similarity/adjacency values for input genomes and perform hierarchical clustering based on the presence or absence of specific k-mers across different genomes, making it invaluable for pangenome studies. It allows researchers to identify conserved and unique genetic elements among a group of organisms, contributing to our understanding of evolutionary relationships, gene function, and the impact of genetic diversity on phenotypic traits [12].

To detect the presence of *P. teres* in planta, different DNA-based markers were developed, such as RAPD (Random Amplified Polymorphism DNA) [13,14,15,16], RFLP (Restriction Fragment Length Polymorphism) [17], AFLP (Amplified Fragment Length Polymorphism) [18,19], and SSR (Simple Sequence Repeat) markers [20]. Different primers have been established for *P. teres* detection using PCR or quantitative real-time PCR (qPCR) [21,22,23,24,25]. Unfortunately, most of these methods used artificial inoculation or were carried out after symptom development on symptomatic material.

To optimize disease management, early detection of the pathogen before symptom apparition is of great importance. In recent years, the use of digital droplet PCR (ddPCR) in plant pathology has grown considerably [26,27]. ddPCR is a robust and sensitive technique that is highly valuable for monitoring low-titer pathogens in complex samples. In cereals, ddPCR has been used to detect different pathogens, including *Tilletia laevis* [28,29], *Fusarium* sp. [30,31], and *Ramularia collo-cygni* [32].

This study aims to develop molecular tools able to detect both forms of *P. teres* in barley leaves before symptoms appear. To achieve this, we delved into the k-mer landscapes of all available *P. teres* genomes to identify a universal consensus sequence. This consensus sequence was then successfully used to design primers and a probe for qPCR and ddPCR assays for early fungal detection.

## 2. Results and Discussion

### 2.1. Primers and Probe Design for P. teres Detection Using a K-Mer Approach

To detect *P. teres*, “consensus” primers and probes were defined using a workflow described in Appendix A. As shown in Figure 1a, the total number of k-mers increased rapidly, reaching approximately 2.2 × 10^8^, while core k-mers—those conserved across all genomes—plateaued at 3.5 × 10^7^. This indicates that a small subset of the genome is highly conserved across all strains. The UpSet plot in Figure 1b highlights the shared k-mer distribution among the reference strains W11 (*Ptt*), SG1 (*Ptm*), and the hybrid strain WAC10721. W11 and WAC10721 share a significant portion of k-mers (around 2.6 × 10^7^), reflecting their close genetic relationship, while SG1 exhibits fewer shared k-mers with the other two strains, indicating divergence in its accessory genome.

The Jaccard similarity heatmap (Figure 1c) reveals three major clusters: Cluster 1 (blue), comprising strains such as WAC10721, FGOB10Pt-1m, and *Ptm* strain SG1; Cluster 2 (blue-green), consisting of *Ptt* strains like NB85, NB73, and NB29; and Cluster 3 (green), which includes *Ptt* strains such as W1-1 and NB85, showing significant overlap within their k-mer similarities. Notably, the hybrid strain WAC10721 appears in Cluster 1, sharing more k-mers with *Ptm* strains but still retaining some overlap with *Ptt* strains, reflecting its mixed genomic background. These findings suggest the presence of conserved core sequences and strain-specific accessory regions, with clear groupings based on Jaccard similarity. The k-mer analysis performed across 22 *P. teres* strains, as depicted in Figure 1, underscores significant genomic variability, particularly in the core regions of the genome. This variability is consistent with broader studies on fungal population genomics, where PanKmer and k-mer-based analyses have been instrumental in identifying both conserved core genomic regions and highly variable accessory regions. For instance, Shi et al. [33] demonstrated that accessory regions, driven by evolutionary pressures such as Horizontal Gene Transfer (HGT), contribute significantly to genomic diversity within fungal populations [34]. Moreover, the divergence observed in the UpSet plot, particularly between the reference strains SG1 and WAC10721, highlights how accessory regions can differ dramatically even among closely related strains, a pattern that reflects the findings of Syme et al. [34], where diversity analyses of *Pyrenophora teres* strains revealed significant divergence in accessory regions due to genomic expansions and rearrangements. Finally, the clustering of strains in the Jaccard similarity heatmap aligns with studies like those by Barragan et al. [35], where hybrid strains exhibit genomic features from both parental lineages, reinforcing their adaptive potential through a mix of core and accessory genomic traits. Several studies report the significance of chromosomal variation in the species, particularly regarding virulence and genetic diversity. Research by Wyatt et al. [36] highlights the involvement of chromosome 2 in significant genetic rearrangements, contributing to variability in pathogenicity and adaptation mechanisms. Additionally, studies focusing on recombination events and chromosomal structural changes further reinforce the idea that chromosome 2 serves as a hotspot for diversity. For example, the work of Ellwood et al. [37] reveals that chromosome 2 in *P. teres* plays a critical role in virulence evolution, exhibiting large-scale genetic exchanges which support the variability in accessory regions.

The pangenome assembly for chromosome 2 was constructed and analyzed to capture both core and accessory genomic elements across the 22 strains of *P. teres*, with particular focus on the reference strains SG1 (*Ptm*) and W11 (*Ptt*). The assembly revealed a high degree of conservation, particularly in genomic regions spanning up to 700 kb, which were consistently present across SG1, W11, and the hybrid strain WAC10721. As shown in Figure 2a, the global structure of the pangenome demonstrates well-conserved core blocks shared across the strains, forming the basis for consensus primer development. These core sequences, especially between SG1 and W11, exhibit substantial stability, making them ideal for downstream molecular applications. The overall assembly was highly contiguous, comprising 397,002 scaffolds with no gaps detected. The N50 scaffold length of 1672 bp and a GC content of 41.68% further reflected the assembly’s robustness and completeness. Additionally, the high-quality reference genomes of SG1 and W11 validate the integrity of the pangenome, ensuring that the core regions of chromosome 2 are well-represented.

In Figure 2b, the alignment of core sequences and variable accessory elements is depicted, with the core regions (shown as dark, aligned blocks) demonstrating consistent conservation across the strains. In contrast, accessory regions exhibit some divergence, particularly in strain-specific elements, which are less conserved and scattered across the pangenome. The conserved regions, extending hundreds of kilobases, are of particular importance, as they highlight critical genomic elements for functional studies. These conserved blocks provide a solid foundation for identifying universal genomic targets, while the accessory regions present opportunities to study strain-specific adaptations and evolutionary divergence. The pangenome structure, optimized through Smoothxg and visualized with ODGI, showcases well-defined connections between the conserved core regions, with accessory sequences diverging into looped or variable paths.

The overall alignment and structure reveal that strains like W11, SG1, and WAC10721 share a substantial portion of their core genome, reinforcing the conserved nature of chromosome 2 across both *Ptt*, *Ptm*, and hybrid forms. This analysis of chromosome 2 underscores its potential as a key genomic region for molecular diagnostics and further functional analyses, particularly for the development of consensus primers based on the identified conserved sequences. The pangenome assembly for chromosome 2 of *P. teres* highlights the remarkable conservation of core genomic regions across multiple strains, including reference strains SG1, W11, and the hybrid WAC10721. These core regions, spanning hundreds of kilobases, are critical for maintaining fundamental biological functions and provide a stable foundation for molecular diagnostics, as well as the development of consensus primers for future studies. Studies such as Wyatt et al. [36] emphasize that these core regions are highly conserved across the genome, particularly in sub-telomeric regions, which are vital for maintaining structural integrity and chromosomal stability. These stable, conserved blocks are ideal for further molecular and functional studies, enabling researchers to target universal sequences that are shared across different strains and forms of *P. teres.* The robustness of these core sequences also ensures the accuracy of pangenome assembly and provides insights into evolutionary stability across the species.

The candidate sequence extracted from chromosome 2 was aligned using Multiple Sequence Alignment (MSA) across 22 *P. teres* strains, including W11 (*Ptt*), SG1 (*Ptm*), and WAC10721 (hybrid) (Appendix A). The alignment demonstrates high conservation of the sequence across the isolates, with minimal observed variability. Key regions of the sequence are consistently aligned across all isolates, confirming their presence in the core genomes of both *Ptt* and *Ptm* forms, as well as in the hybrid strains. The results reveal strong conservation across core genomes, consistent with findings from similar studies that emphasize the stability of core genomic regions in *P. teres* [36]. The k-mer 18 analysis, which detects repetitive elements within these conserved regions, aligns with previous research showing that repetitive sequences are common in fungal genomes yet do not disrupt overall genomic stability [38]. This observation sets the stage for future confirmatory studies, such as ddPCR assays, to validate the presence and quantify these high-copy-number k-mers, thus contributing to a more granular understanding of the genomic structure and diversity of *P. teres*.

Moreover, further annotation of the sequence was performed using Pfam and AntiSMASH databases (Appendix A). The annotation identified several functional domains within the sequence, including cytochrome P450 (PF00067), short-chain dehydrogenase/reductase (SDR) (PF00106), and major facilitator superfamily (MFS) transporters (PF07690), all of which are involved in various biosynthetic processes. Additionally, non-ribosomal peptide synthetase (NRPS) and polyketide synthase (PKS) domains were detected, indicating the sequence’s involvement in secondary metabolite production. These annotations suggest a wide functional range of the sequence across the analyzed genomes. The functional domains identified, such as cytochrome P450, SDR, and MFS transporters, are typical of fungal metabolic pathways and have been associated with essential biosynthetic processes [39]. Furthermore, the detection of NRPS and PKS domains, known to drive secondary metabolite production, supports the hypothesis that these conserved sequences play a crucial role in secondary metabolism in *P. teres*.

The pank-mer analysis carried out on the 22 *P. teres* strains was used to define universal primers, named as PT_Cons1 F (Forward)/PT_Cons1 R (Reverse) and a TaqMan probe, called FAM-PT_Cons1 probe M C8E O (Cons1 for Consensus). The results (Appendix A) indicate that the sequences obtained from the k-mer analysis exhibit a high degree of similarity across the entire collection of *P. teres* genomes. The consistency in these sequences proposes their potential as universal markers for any environmental *P. teres* strain. This is particularly significant given the limited resolution of ITS barcoding in differentiating between *P. teres* strains, where ITS sequences may not provide enough discriminatory power to distinguish closely related strains. Additionally, the alignment (Appendix A) highlights the presence of conserved regions within the sequences, annotated by the forward/reverse primers and probe. This indicates that these regions are stable across different genomes and may be valuable for developing universal detection assays.

K-mer analysis has been previously applied in several “omic” analyses, like genome characterization or assemblies, and metagenomic analyses [40,41,42], but, to our knowledge, our study has developed this approach for the first time to define consensus primers that detect both *Ptt* and *Ptm* strains. The pankmer, also known as the k-mer pangenome, is a concept used in bioinformatics and genomics. In this context, pan k-mer analysis can identify fundamental genomic elements shared by all members of a species, while also considering accessory or variable k-mers present only in specific strains. This analysis offers insights into genetic diversity and commonalities within a population or species. Given that *Ptt* and *Ptm* are genetically distinct forms considered as two species [23,38], this k-mer approach appears to be pertinent and allows us to define consensus primers and probes based on the 22 fungal genomes available.

### 2.2. Efficiency, Specificity, and Limit of Detection (LOD) of Primers/Probe Designed

Field sampling of *Hordeum vulgare* across diverse Moroccan agro-ecological zones revealed varying infection responses to *Pyrenophora teres*, as evidenced by molecular confirmation and infection response ratings based on [43,44]. The infection responses ranged from moderate (4–4.5) to highly virulent (7–8), indicating a significant presence of pathogenic forms within the sampled regions (Appendix A). The high similarity of these conserved sequences across both *P. teres* f. *teres* (*Ptt*) and *P. teres* f. *maculata* (*Ptm*) forms, as reflected in the phylogenetic tree, underscores the genetic closeness of the Moroccan strains (Appendix A).

First, to validate the designed primers, we tested them against two strains of *Ptt* and two *Ptm* (Appendix A). Amplification and melting curves are presented in Appendix A. The efficiency of primers without and with a probe was tested on one *Ptt* and one *Ptm* strain using a dilution from a PCR product (Appendix A). The remainder of this study focused on the use of the probe.

To test the probe specificity, common barley pathogens retrieved in the barley phyllosphere [2], such as *Rhynchsoporium* sp. (scald disease), *Cochliobolus sativus* (formerly *bipolaris sorokiniana*, causing spot blotch), and *Fusarium oxysporum*, as well as in *Penicillium hordei* found in barley seeds [45,46], were also used. The results presented in Figure 3 demonstrate the specificity of the probe.

To determine the limit of detection (LoD), the probe was tested on a 10-fold serial dilution of pure *P. teres* f. *teres Ptt* (*Ptt*) or *P. teres* f. *maculata* (*Ptm*) gDNA (50 ng to 0.05 pg). The standard curve for *P. teres* was generated by plotting the log of DNA (ng) against the Cq value determined using qPCR. The recovered curves showed excellent linearity of amplification for both *Ptt* and *Ptm* (Figure 4). The qPCR efficiency values retrieved from these standard curves were 101.6 for *Ptt* and 101.3% for *Ptm*, meeting the requirement set in the MIQE guidelines [47]. Notably, we observed a sensitivity of 0.05 ng for both *Ptt* and *Ptm*, which is comparable to the sensitivity observed for other fungal phytopathogens [48].

### 2.3. Detection of Ptt and Ptm Using ddPCR Assay Before Symptoms Appear

Digital droplet PCR (ddPCR) offers a number of advantages over qPCR [27,49], including better sensitivity and reduced susceptibility to inhibition due to microdroplet fractionation [50,51]. These advantages have contributed to a significant increase in the use of this technique for in planta detection of pathogenic microorganisms in recent years [26].

In our study, we demonstrated that ddPCR is five times more sensitive than qPCR, capable of detecting as low as 0.01 ng of gDNA (Figure 5). This result is in accordance with previous studies carried out on different pathosystems which show that ddPCR exhibits greater sensitivity compared to qPCR [29,52,53,54].

A previous study on barley infected with *D. teres* demonstrated that fungal detection using qPCR was only possible after the onset of symptoms [55]. To investigate whether *P. teres* could be detected before symptom apparition, we conducted a study on the kinetics of barley leaf infection. The barley genotype Oussama was infected with the *Ptt3* or *Ptm6* strain, isolated from fields (Appendix A), and detection was performed on infected leaves at 0, 2, 4, and 10 days post-infection (dpi). First, specificity was tested using non-infected barley leaves and NTC (water) as negative controls, while gDNA from fungal strains served as positive controls (Appendix A). Moreover, by testing 30 “blank” samples (NTC), the LoB was determined to be 1.77 copies/reaction.

As expected and shown in Figure 6a (and Appendix A), negative controls, including non-infected barley leaves (Control) and NTC, were below the LoB. Positive controls for *Ptt* and *Ptm* were detected, respectively, at 6.87 and 3.89 copies/reaction, respectively. For *in planta* detection, both *Ptt* and *Ptm* were detected as early as 0 dpi and reached an optimum at 10 dpi with respective values of 11.9 and 50.8 copies/reaction.

In terms of symptom development, we observed that the asymptomatic phase in barley leaves was shorter for *Ptm* (2 days) compared to *Ptt* (4 days) (Figure 6b). These results align with previous observations [6,56]. Such a difference could be explained by a difference in lifestyle between the two forms, *Ptt* and *Ptm*. Specifically, *Ptt* behaves as a necrotroph, while *Ptm* acts as a hemibiotroph [57]. Furthermore, *Ptm* exhibits significantly higher necrotrophic and saprotrophic growth rates than *Ptt* [58].

## 3. Conclusions

Different molecular tools based on classical PCR or qPCR have previously been developed to target *P. teres* [15,21,22,25]. In the last few years, ddPCR technology has emerged in the field of plant pathology [26]; however, it has only been applied in one study on barley to detect *Ramularia collo-cygni* [32]. In this study, a K-mer approach was used to define consensus primers and a probe based on the 22 available *P. teres* fungal genomes. These primers and the probe enabled the detection of both forms of *P. teres* f. *teres* and *P. teres* f. *maculata*, which cause the net-form and spot-form in barley, respectively, even before symptom development. Thus, we have developed universal markers for the detection of both forms of *P. teres*, effectively overcoming environmental variations. For further investigations, it would be interesting to validate these tools on field leaves.

## 4. Materials and Methods

### 4.1. Microorganisms: Isolation of Moroccan Fungal Strains

Fungal specimens were collected from the ICARDA-INRA research farm during the advanced stages of stem growth following a workflow described in Appendix A. Isolates of *Ptm* and *Ptt* were derived from mature barley foliage exhibiting distinct disease symptoms. These specimens were air-dried, stored at cool temperatures, and subjected to meticulous surface sterilization using a sequence of chemical treatments before rinsing with sterile water. The cleansed leaf fragments were then dried and placed in humidified conditions to encourage fungal growth, with exposure to a cycle of light and dark periods. Post-incubation, the strains were cultured on PDA media for several weeks at refrigerated temperatures. For longevity, the fungal material was cryopreserved at −80 °C. The identification of the strains was further confirmed using molecular analysis, specifically using the internal transcribed spacer (ITS) region (Appendix A).

### 4.2. Plant Infection

Plant infection was realized as described previously [55]. Briefly, barley seeds (Moroccan cultivar “Oussama”) were sown in plastic pots containing 90 ± 5 g of substrate GramoFlor. For each pot, ten seeds were sown. Seedlings were grown to the two-leaf stage (10 days) under controlled conditions in incubators (Aralab, Rio de Mouro, Portugal), following a 23 °C/22 °C day/night temperature cycle, 80% relative humidity, a 14 h/10 h day/night photoperiod. At ten days post-sowing, each leaf was infected with 1 mL of *P. teres* spore suspension at 3000 conidia/mL using a handheld sprayer until runoff. Barley leaves were collected at 0, 2, 4, 7, and 10 days post-infection (dpi), with non-infected barley leaves serving as controls. For the molecular experiments, i.e., qPCR and ddPCR, the leaves were harvested, fixed immediately in liquid nitrogen, and stored at −80 °C until DNA extraction.

### 4.3. DNA Extraction

Mycelium from a pure fungal strain cultured on Petri dishes was harvested using a sterile P1000 cone and ground in liquid nitrogen. Infected and control (non-infected) barley leaves were similarly ground in liquid nitrogen. Genomic DNA was extracted from 200 mg powder using the CTAB method according to Stefanova et al. (2013) [59]. The DNA concentration and quality of the DNA samples were measured using a Nanodrop One Microvolume spectrophotometers (Thermo Scientific, Waltham, MA, USA) and visualized using migration in agarose gel. Extracted DNA was stored at −20 °C until further use.

### 4.4. Primers/Probe Design Using K-Mer Analysis

To define consensus primers capable of detecting *P. teres* f. *teres* (*Ptt*), *P. teres* f. *maculata* (*Ptm*), and hybrid forms of *P. teres*, we utilized the genomic sequences of 22 available strains (Appendix A). A k-mer analysis was performed to identify candidate primer and probe sequences, following the workflow outlined in Appendix A. The Pankmer software v.0.20.4 [12] was employed to construct a pan-genome k-mer set with a k-mer length of 18 nucleotides, balancing specificity and computational efficiency given the genomic complexity of *P. teres*. Using Pankmer, we counted k-mers across the 22 genomes, which formed the basis for constructing an adjacency matrix. This matrix enabled the identification of core k-mers common to all genomes with 18 k-mer, and their distribution was assessed using the Jaccard similarity metric for clustering. An UpSet plot was generated to visualize the intersection of core k-mers among the 22 strains representing *Ptt*, *Ptm*, and hybrid forms. Next, Unikmer (https://github.com/shenwei356/unikmer, accessed on 4 January 2023) was employed to extract sequences common to the core k-mers on chromosome 2, further delineating shared genetic elements associated with chromosome 2 and contig-level data, which were essential for downstream analysis. The pangenome was constructed using the nf-core/pangenome pipeline (v1.1.2-gaf6d1dd) and was executed via Nextflow v24.04.4 [60,61]. Core k-mers from 18 haplotypes were used as input for this process. Sequence alignment was performed using Wfmash v0.19.0 (https://github.com/waveygang/wfmash, accessed on 6 June 2023) with a sequence identity threshold of 90%, and the alignments were processed through Seqwish v0.7.10 (https://github.com/ekg/seqwish, accessed on 6 June 2023) to generate the pangenome graph using sparse factor randomization. Graph smoothing was carried out with Smoothxg v0.7.4. (“https://github.com/pangenome/smoothxg”, accessed on 6 June 2023), applying predefined sequence length parameters (700, 900, 1100). Following this, layout optimization and visualization were performed with ODGI v0.8.6. (“https://github.com/pangenome/odgi”, accessed on 6 June 2023). The pangenome graph was first constructed using an odgi build, optimized with odgi layout and odgi sort, and finally visualized with odgi viz. After constructing the pangenome, Unikmer was once again used to extract specific genomic regions from chromosome 2 that were conserved across the isolates with up to 700 kb. To further enhance our understanding of these regions, AntiSMASH [62] was employed to investigate secondary metabolite clusters. The combined results from the database Pfam and AntiSMASH allowed us to precisely identify genomic regions associated with metabolite production, guiding the selection of target sequences for primer design [62,63].

For primer and probe design, the sequences from the 22 genomes of *P. teres* were filtered to include only those up to 700 bp in length. This selection criterion was applied to capture the most relevant genomic segments for effective primer targeting. Subsequently, a local BLAST search was conducted on these filtered sequences to identify highly conserved regions. Following this, primers and probes were designed with a target amplicon size of 75–200 bp to ensure optimal amplification efficiency. Primers were designed to be 18–30 bases in length, with a GC content of 40–60%, aiming for a melting temperature (Tm) between 50 and 65 °C. A maximum ΔTm of less than 3 °C between the forward and reverse primers was set. The probe was strategically selected to lie between the two primers without overlap, with a Tm 5–10 °C higher than that of the primers. The probe was designed to be shorter than 30 nucleotides to maintain a stable baseline signal intensity during ddPCR. Primers and probe details (sequences, amplicon length, amplicon Tm) are provided in Appendix A.

### 4.5. Specificity

Primers and probe specificity were tested against DNA extracted from two environmental *P. teres* strains (one *Ptt* and one *Ptm*) and other barley pathogens from the BCCM/MUCL collection: *Rhynchsoporium* sp. MUCL 30115, *Cochliobolus sativus* MUCL 504, *Penicillium hordei* MUCL 39559, and *Fusarium oxysporum* MUCL 520. The experiment was conducted with three technical replicates.

### 4.6. qPCR Assay

qPCR experiments were carried out according to MIQE guidelines [47]. Quantitative real-time PCRs were performed using qPCRBIO SyGreen Blue Mix Lo-Rox (Eurobio Scientific, Les Ulis, France) or 2XPCRBIO Taq Mix red (PCR BioSystems, London, UK) in white 384-well plates (Brand, Wertheim, Germany) on a CFX Opus 384 instrument (Bio-Rad, Hercules, CA, USA). The cycle threshold (Ct) of each amplification curve was calculated using the second derivative maximum method in CFX Maestro™ Software v1.1. PCR conditions (without the probe) were 95 °C for 3 min, followed by 40 cycles of 5 s at 95 °C and 30 s at 60 °C. For each reaction, a single amplicon with the expected melting temperature was obtained. When the probe was used, the conditions were 95 °C for 10 min followed by 40 cycles of 15 s at 95 °C and 1 min at 60 °C.

### 4.7. ddPCR Assay

ddPCR experiments were carried out according to digital MIQE guidelines [64]. Digital droplets PCR was performed using ddPCR SuperMix for Probes (no dUTP) (Bio-Rad) in DG32 Cartridges and clear ddPCR 96-well plates (Bio-Rad) using the Automated Droplet Generator, C1000 Touch Thermal Cycler, and QX200 Droplet Reader (Bio-Rad).

In planta fungal quantification was performed on 10 ng of purified gDNA and in a 22 μL reaction volume. The reaction mixture comprised 11 μL of Supermix, 600 nM of each PT_Cons1-F/R primer, 400 nM of the PT_Cons1probe, and 1 μL of the DNA template. Aliquots (20 μL) from each reaction mixture were loaded into the sample wells of a DG32 cartridge, and 70 μL of Droplet Generation Oil was added to each of the corresponding oil wells to generate microdroplets, which were then transferred to a 96-well PCR plate. After heat-sealing the plate with pierceable foil using the PX1™ PCR Plate Sealer (Bio-Rad), the PCR amplifications were with an initial step of 95 °C for 10 min, followed by 45 cycles at 95 °C for 30 s, 60 °C for 1 min, with thermal ramp of 2 °C/s. Next, the microdroplets from each well were read individually using the QX200™ Droplet Reader, and the data were analyzed using QuantaSoft™ Analysis Pro Software v1.0.596 (Bio-Rad, Hercules, CA, USA). Positive microdroplets containing the PCR-amplified products were discriminated from the negative ones by applying a fluorescence amplitude threshold at the highest point of the negative microdroplet cluster. Reactions with more than 10,000 accepted microdroplets per well were used for the analysis. The starting target DNA concentration of the target nucleic acids within each sample was calculated based on the ratio of positive to total microdroplets using Poisson statistics.

To establish the limit of blank (LOB) with a 95% confidence level, at least 30 ddPCR assays were performed on blank samples that contained no sequence for any given target but had identical backgrounds to kinetic infection samples. The LOB was calculated following the Clinical and Laboratory Standards Institute (CLSI) EP17-A2 standard (Protocols for Determination of Limits of Detection and Limits of Quantitation; Approved Guideline).

### 4.8. Data Analysis

PanKmer’s pankmer collect was used to calculate and plot collection curves, assessing the pangenome completeness and core k-mers among the 22 genomes. An UpSet plot was generated to visualize k-mer intersections, and MATLAB v 3.8.0 was used for the visualization of the adjacency matrix [65]. The ggmsa package in R was used to visualize the consensus sequence and its alignment across the genome collection using the ClustalW algorithm implemented in the MSA (Multiple Sequence Alignment) package [66,67].

## Figures and Tables

**Figure 1 ijms-25-11980-f001:**
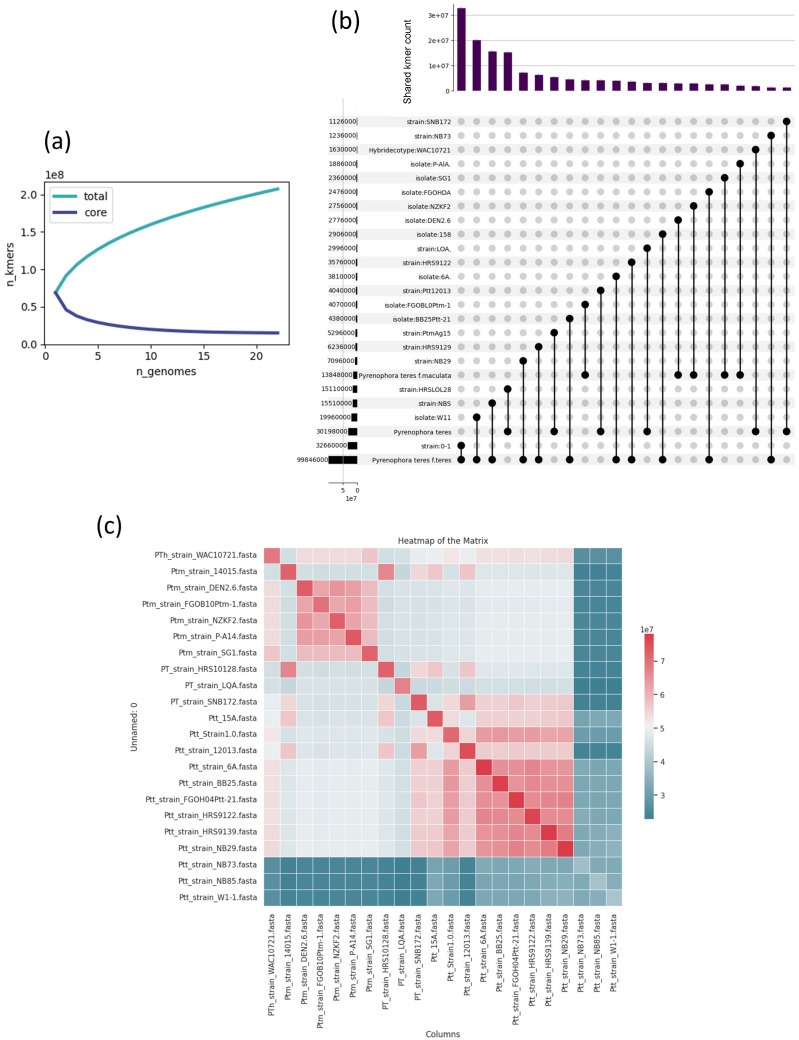
Integrated pangenomic analysis of *Pyrenophora teres*. (**a**) Collection curve depicting the accumulation of total and core k-mers as more genomes are sequenced, being the 22 genomes, suggesting a comprehensive pangenome that is not open, but rather is interconnected among different forms. (**b**) UpSet plot illustrating the intersections of k-mers for three representative groups of *P. teres*—‘*Ptt* strain W1-1’, ‘*Ptm* strain SG1’, and ‘hybrid strain WAC10721’—showing shared and unique genomic elements. (**c**) A heatmap of the adjacency matrix displaying k-mer-based clustering of various *P. teres* strains, offering insights into the relationships both within and between *P. teres* forms.

**Figure 2 ijms-25-11980-f002:**
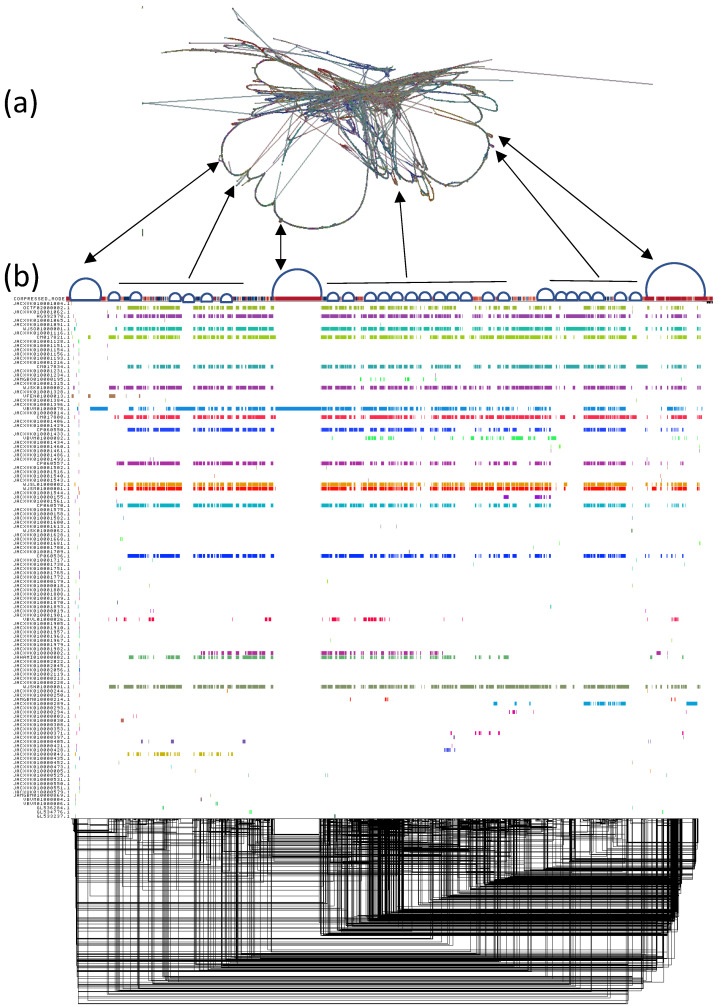
Pangenome graph visualization of *P. teres* Chromosome 2. (**a**) Pangenome graph showing the structural relationships and conserved regions across 22 *P. teres* strains. Core conserved regions are depicted with tightly aligned paths, while the looping and branching paths represent variable or accessory regions unique to specific strains. (**b**) Linear visualization of the pangenome alignment, showing core genome regions in consistent colored blocks shared across multiple strains.

**Figure 3 ijms-25-11980-f003:**
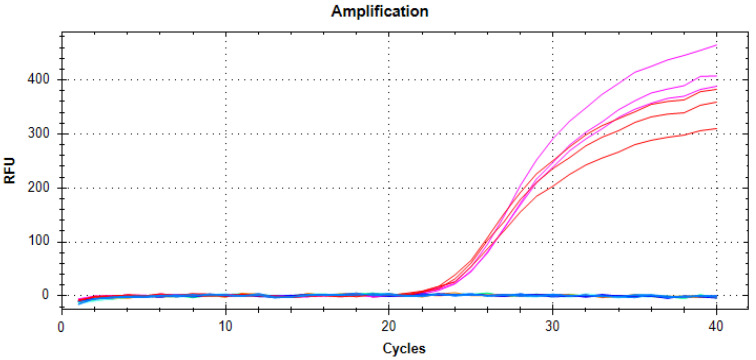
Probe specificity tested against genomic DNA (10 ng) extracted from *P. teres* f. *teres* (in pink, three repetitions) or f. *maculata* (in red, three repetitions), and non-*P. teres* species: *Cochliobolus sativus* = *bipolaris sorokiniana*, *Fusarium oxysporum*, *Penicillium hordei*, and *Rhynchsoporium secalis*. A non-template control (NTC) was used as the negative control.

**Figure 4 ijms-25-11980-f004:**
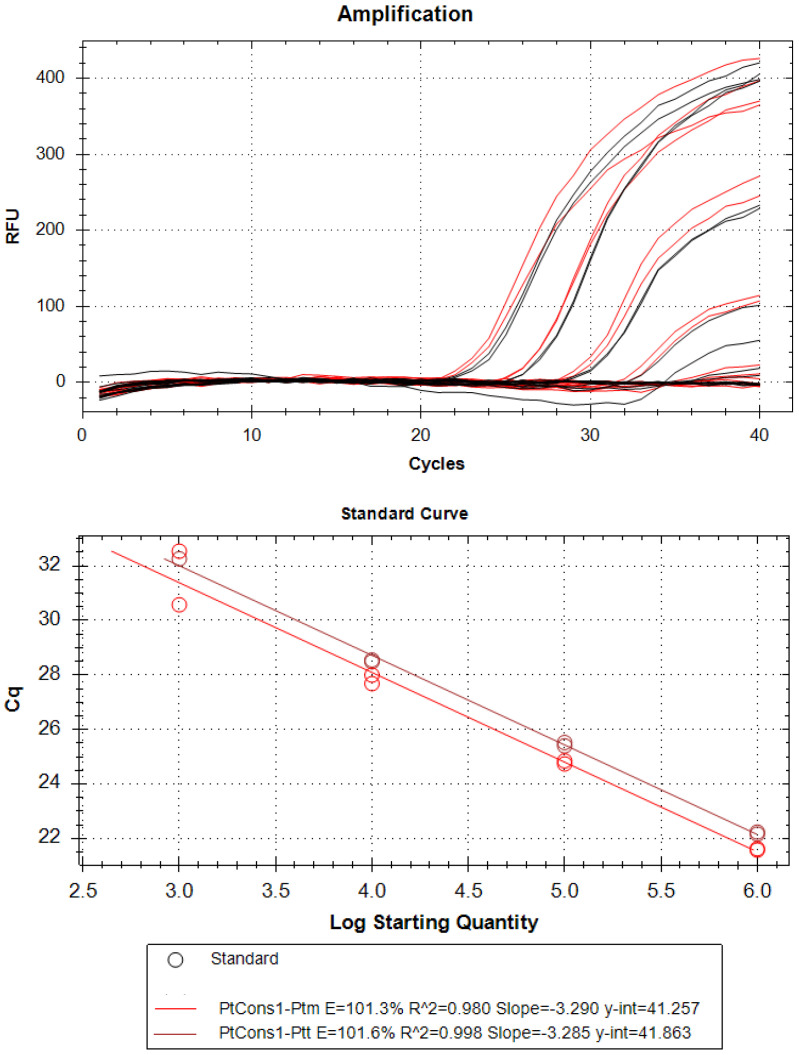
Efficiency of probe using the templates of *Ptt* or *Ptm* genomic DNA (50 ng–0.05 pg).

**Figure 5 ijms-25-11980-f005:**
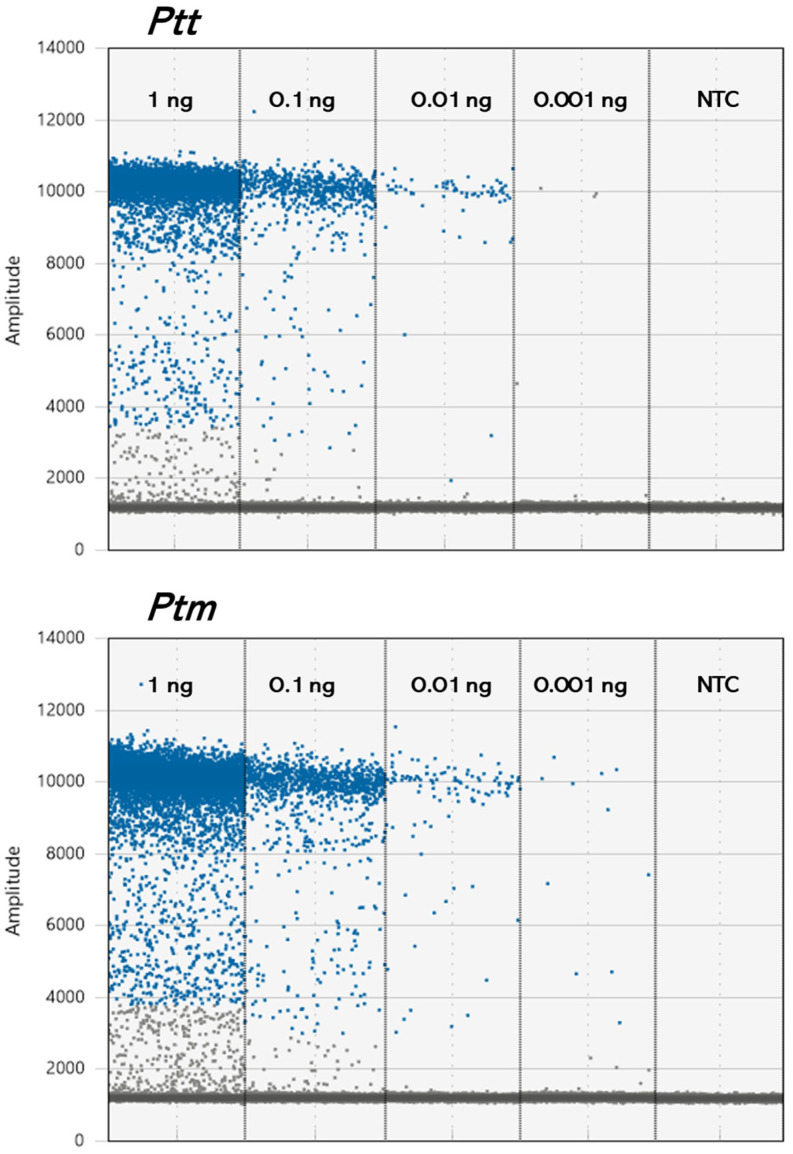
Limit of detection using the templates of *Ptt* or *Ptm* genomic DNA.

**Figure 6 ijms-25-11980-f006:**
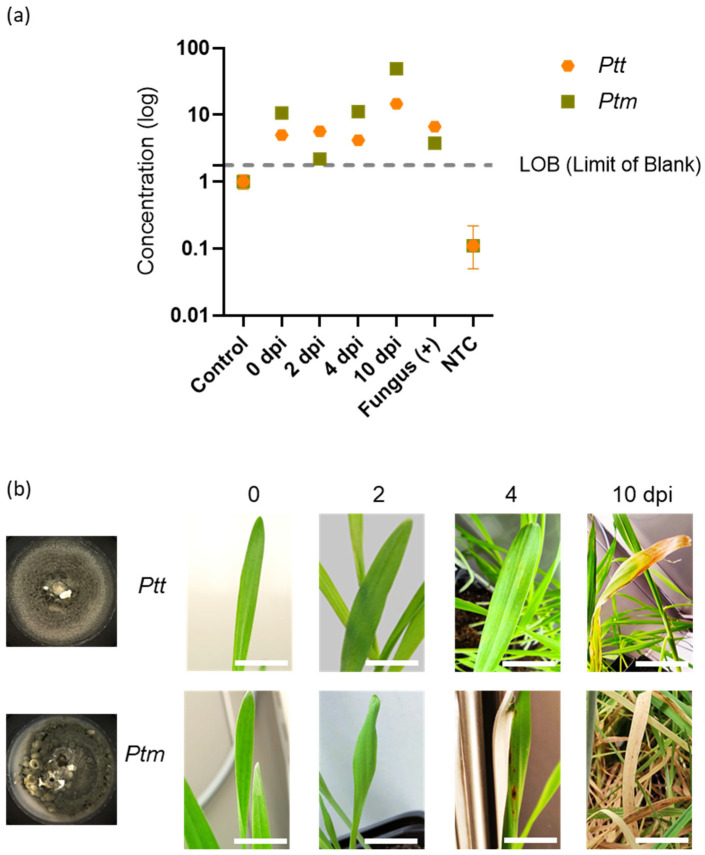
Detection of both *Ptt* and *Ptm* strains using ddPCR (**a**) before symptom apparition (**b**) at 0, 2, 4, and 10 days after infection. For ddPCR, control (non-infected plants) and NTC (water) were used as negative controls. gDNA from fungal strains served as positive controls (Fungus (+)). An in planta detection of *Ptt* or *Ptm* was realized at 0, 2, 4, and 10 days after infection. The results presented are from one representative experiment among three independent biological repetitions. Scale bar = 2 cm.

## Data Availability

The raw data supporting the conclusions of this article will be made available by the authors on request.

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
