# Peer review of "Early Detection of Both Pyrenophora teres f. teres and f. maculata in Asymptomatic Barley Leaves Using Digital Droplet PCR (ddPCR)"

_ijms, 2024, doi:10.3390/ijms252211980_

Round 1

Reviewer 1 Report

Comments and Suggestions for Authors

Dear authors,

The original paper entitled “Early detection of both Pyrenophora teres f. teres and f. maculata in asymptomatic barley leaves by digital droplet PCR (ddPCR)” is relatively well-written, structured and developed by Bouhouch et al. in suitable English with a clear structure. They developed and evaluated molecular techniques and PCR-based methods, including ddPCR, qPCR, and TaqMan probe assays for early detection of both Ptt and Ptm. This study was interesting; however, some points of view need to be addressed. After major revisions, this manuscript must be reviewed again for evaluation.

1.     Analytical and diagnostic sensitivity and specificity coefficients are used to evaluate the developed diagnostic methods. Obviously, you considered the analytical sensitivity and specificity for this evaluation. You also need to calculate and discuss the diagnostic sensitivity and specificity of the methodologies you developed in your study. Please summarize all of them in an evaluation table.

2.     Also, you need to calculate the reproducibility coefficient in your study and discuss the data in the evaluation table.

3.     Some figures (Figures 4 and 5) are not illustrated very professionally and are unsuitable for publishing in IJMS. You may visualize these data again.

4.     Microbial strain names should be in italic form throughout the manuscript. Please double-check all of them.

5.     Figure 3 might be regarded as the supplementary data.

6.     Figure 1 is not clear. Please add a higher-quality version of this figure to the manuscript.

7.     Regarding the qPCR method, please add the amplification and melting curves of the diluted samples and discuss them in the manuscript. The methodology of this method in your study is not very clear. 

Author Response

Reviewer 1

The original paper entitled “Early detection of both Pyrenophora teres f. teres and f. maculata in asymptomatic barley leaves by digital droplet PCR (ddPCR)” is relatively well-written, structured and developed by Bouhouch et al. in suitable English with a clear structure. They developed and evaluated molecular techniques and PCR-based methods, including ddPCR, qPCR, and TaqMan probe assays for early detection of both Ptt and Ptm. This study was interesting; however, some points of view need to be addressed. After major revisions, this manuscript must be reviewed again for evaluation.

Comment 1.     Analytical and diagnostic sensitivity and specificity coefficients are used to evaluate the developed diagnostic methods. Obviously, you considered the analytical sensitivity and specificity for this evaluation. You also need to calculate and discuss the diagnostic sensitivity and specificity of the methodologies you developed in your study. Please summarize all of them in an evaluation table 

Comment 2.     Also, you need to calculate the reproducibility coefficient in your study and discuss the data in the evaluation table.

Responses to comments 1 and 2: The main objective of our study was not to compare qPCR and ddPCR, that is why we didn’t include table comparing the two methods. However, for greater clarity we have added details in the text and in supplementary data, in the hope that this will meet your expectations.

Comment 3.     Some figures (Figures 4 and 5) are not illustrated very professionally and are unsuitable for publishing in IJMS. You may visualize these data again.

Response to comment 3: OK, changed

Comment 4.     Microbial strain names should be in italic form throughout the manuscript. Please double-check all of them.

Response to comment 4: OK, changed

Comment 5.     Figure 3 might be regarded as the supplementary data.

Response to comment 5: OK, moved in Supplementary data (now, Supplementary figure S2)

Comment 6.     Figure 1 is not clear. Please add a higher-quality version of this figure to the manuscript.

Response to comment 6: OK, changed

Comment 7.     Regarding the qPCR method, please add the amplification and melting curves of the diluted samples and discuss them in the manuscript. The methodology of this method in your study is not very clear.

Response to comment 7: This part was changed for greater clarity. We added, as you asked, the amplification and melting curves and figure on primers efficiency without and with probe in supplementary data.

Reviewer 2 Report

Comments and Suggestions for Authors

In this manuscript, a K-mer approach was used to design consensus primers and a probe based on the 22 available P. teres fungal genomes. The authors declaimed that these primers and probe enabled the detection of both forms of P. teres f. teres and P. teres f. maculata, which cause the net-form and spot-form net blotch on barley leaves, respectively, even before symptoms development. However, there are several issues need to be solved in this manuscript.

1. In Introduction section (line35-40), specific terms such as Pyrenophora teres, Pyrenophora graminea, Fusarium sp., Rynchosporium 35 commune, Puccinia hordei, and Cochliobolus sativus” in the manuscripts need to be italicized as Pyrenophora teres, Pyrenophora graminea, Fusarium sp., Rynchosporium 35 commune, Puccinia hordei, and Cochliobolus sativus”. Please check the full context.

2. In Keywords: barley, …”. The comma ,” here needs to be changed into a semicolon ( ; ”).

3. The images in figure 1 and 2 are not of high resolution, which need to be replaced by higher resolution ones, even vector graphics.

4. In line 85-89, is there any mistake for these phrases “reaching approximately 2.2 × 108”, “plateaued at 3.5 × 107” and “around 2.6 × 107”? They seem to be “reaching approximately 2.2 × 108”, “plateaued at 3.5 × 107” and “around 2.6 × 107”, respectively.

5. The “Results and discussion” section is suggested to separate into “Results section” and “discussion section”.

6. In Figure 7. “Detection of both Ptt and Ptm strains by ddPCR before symptom apparition.” should be “Detection of both Ptt and Ptm strains by ddPCR before symptom apparition.”. And the authors should give more details about panel “a” and “b” in the figure legend.

Comments on the Quality of English Language

In this manuscript, a K-mer approach was used to design consensus primers and a probe based on the 22 available P. teres fungal genomes. The authors declaimed that these primers and probe enabled the detection of both forms of P. teres f. teres and P. teres f. maculata, which cause the net-form and spot-form net blotch on barley leaves, respectively, even before symptoms development. However, there are several issues need to be solved in this manuscript.

1. In Introduction section (line35-40), specific terms such as Pyrenophora teres, Pyrenophora graminea, Fusarium sp., Rynchosporium 35 commune, Puccinia hordei, and Cochliobolus sativus” in the manuscripts need to be italicized as Pyrenophora teres, Pyrenophora graminea, Fusarium sp., Rynchosporium 35 commune, Puccinia hordei, and Cochliobolus sativus”. Please check the full context.

2. In Keywords: barley, …”. The comma ,” here needs to be changed into a semicolon ( ; ”).

3. The images in figure 1 and 2 are not of high resolution, which need to be replaced by higher resolution ones, even vector graphics.

4. In line 85-89, is there any mistake for these phrases “reaching approximately 2.2 × 108”, “plateaued at 3.5 × 107” and “around 2.6 × 107”? They seem to be “reaching approximately 2.2 × 108”, “plateaued at 3.5 × 107” and “around 2.6 × 107”, respectively.

5. The “Results and discussion” section is suggested to separate into “Results section” and “discussion section”.

6. In Figure 7. “Detection of both Ptt and Ptm strains by ddPCR before symptom apparition.” should be “Detection of both Ptt and Ptm strains by ddPCR before symptom apparition.”. And the authors should give more details about panel “a” and “b” in the figure legend.

Author Response

Reviewer 2

In this manuscript, a K-mer approach was used to design consensus primers and a probe based on the 22 available P. teres fungal genomes. The authors declaimed that these primers and probe enabled the detection of both forms of P. teres f. teres and P. teres f. maculata, which cause the net-form and spot-form net blotch on barley leaves, respectively, even before symptoms development. However, there are several issues need to be solved in this manuscript.

Comment 1. In Introduction section (line35-40), specific terms such as “Pyrenophora teres, Pyrenophora graminea, Fusarium sp., Rynchosporium 35 commune, Puccinia hordei, and Cochliobolus sativus” in the manuscripts need to be italicized as “Pyrenophora teres, Pyrenophora graminea, Fusarium sp., Rynchosporium 35 commune, Puccinia hordei, and Cochliobolus sativus”. Please check the full context.

Response to comment 1: OK changed

Comment 2. In Keywords: “ barley, …”. The comma “,” here needs to be changed into a semicolon ( “; ”).

Response to comment 1: OK changed

Comment 3. The images in figure 1 and 2 are not of high resolution, which need to be replaced by higher resolution ones, even vector graphics.

Response to comment 3: OK changed

Comment 4. In line 85-89, is there any mistake for these phrases “reaching approximately 2.2 × 108”, “plateaued at 3.5 × 107” and “around 2.6 × 107”? They seem to be “reaching approximately 2.2 × 108”, “plateaued at 3.5 × 107” and “around 2.6 × 107”, respectively.

Response to comment 4: OK changed

Comment 5. The “Results and discussion” section is suggested to separate into “Results section” and “discussion section”.

Response to comment 5: For easier reading, we prefer to keep the section grouped

Comment 6. In Figure 7. “Detection of both Ptt and Ptm strains by ddPCR before symptom apparition.” should be “Detection of both Ptt and Ptm strains by ddPCR before symptom apparition.”. And the authors should give more details about panel “a” and “b” in the figure legend.

Response to comment 6: OK we added more details (now Figure 6)

Reviewer 3 Report

Comments and Suggestions for Authors

This study developed primers and a TaqMan probe to detect both Ptt and Ptm. I have the following suggestions:

1.      4.1. Microorganisms: isolation of Moroccan fungal strains. The identification of the strains was further confirmed by molecular analysis, specifically using the internal transcribed spacer (ITS) region. Its is enough to distinguish Ptt and Ptm ?

2.      How many isolated Ptt and Ptm were adopted to validate the Primers and probe?

3.      2.3. Detection of Ptt and Ptm by ddPCR assay before symptoms appear. Information about this test should be added in method part.

4.      The title is Early detection of both Pyrenophora teres f. teres and f. maculata 2 in asymptomatic barley leaves by digital droplet PCR. Detection of field barley samples should be tested.

5. pay attention to writing such as Latin name.

Author Response

Reviewer 3

This study developed primers and a TaqMan probe to detect both Ptt and Ptm. I have the following suggestions:

Comment 1.      4.1. Microorganisms: isolation of Moroccan fungal strains. The identification of the strains was further confirmed by molecular analysis, specifically using the internal transcribed spacer (ITS) region. Its is enough to distinguish Ptt and Ptm?

Response to comment 1: The distinction was made firstly on the basis of symptoms observed after infection on the leaves (Supplementary figure S3) and confirmed after by ITS sequencing.

Comment 2.      How many isolated Ptt and Ptm were adopted to validate the Primers and probe?

Response to comment 2: To validate primers, first assay was realized on 2 environmental Ptt strains and 2 environmental Ptm strains from different agrobiological area (we added data in Supp figS5). To validate probe, we used one Ptt and one Ptm (we added data in Supp figS6).

Comment 3.      2.3. Detection of Ptt and Ptm by ddPCR assay before symptoms appear. Information about this test should be added in method part.

Response to comment 3: This is explained in the section 4.2 Plant infection and we added details in ddPCR part.

Comment 4.      The title is Early detection of both Pyrenophora teres f. teres and f. maculata 2 in asymptomatic barley leaves by digital droplet PCR. Detection of field barley samples should be tested.

Response to comment 4: we added a sentence in the conclusion part for future investigations

Comment 5. pay attention to writing such as Latin name.

Response to comment 5: OK, changed

Round 2

Reviewer 1 Report

Comments and Suggestions for Authors

Dear authors,

Thank you very much for revising.

No more comments. 

Comments on the Quality of English Language

Dear Editor,

I have no more comments. This paper can be considered for publishing in its present form. 

Reviewer 2 Report

Comments and Suggestions for Authors

As most of the issues concerned have been revised, I suggest that the manuscript might be accepted.

Reviewer 3 Report

Comments and Suggestions for Authors

This revised version has improved obviously and dissolved my comments. I suggested